## Neglected Tropical Diseases

# Assessment of exposure to zoonoses and perceptions of zoonotic transmission surrounding the Bwindi impenetrable forest, Uganda

Nahabwe Haven [1,2*], Birungi Mutahunga[1], Scott Kellermann[1,3], Jalika Joyner[4], Julia Lippert[2,3], Evan Andrew Rusoja[4,5], Michael Wilkes[4], Gilbert Mateeka[6], Benard Ssebide[7], Charlotte Aguti[1], Isaac Ahwera[1], Charles Mugisa[1], Felista Nanono[6], Prossy Katushabe[6], Nicole R. Gardner[4], Christine Kreuder Johnson[4], Tierra Smiley Evans[2,4*]

**1** Bwindi Community Hospital, Kanungu District, Uganda, **2** Infectious Diseases and Vaccinology Division, Department of Integrative Biology, University of California, Berkeley, California, United States of America, **3** Tulane University, School of Public Health and Tropical Medicine, New Orleans, Louisiana, United States of America, **4** University of California Davis, Davis, California, United States of America, **5** Alameda Health System, Oakland, California, United States of America, **6** Rugarama Hospital, Kabale District, Uganda, **7** Gorilla Doctors, Mountain Gorilla Veterinary Project, Inc., Kampala, Uganda

\* hahotice@googlemail.com (HN); tsmevans@berkeley.edu (TSE)

## Abstract

### Introduction

Emerging infectious diseases with regional spread and potential to escalate to a global pandemic have increased in the last century. Western Uganda has experienced many emerging infectious disease outbreaks over the last five decades, some with worldwide implications. Outbreaks have originated from wild animal reservoir hosts including Marburg and Sudan virus. The goal of this study was to better understand communities contacts with wild and domestic animals and their knowledge of potential disease risks associated with these interactions around Bwindi Impenetrable National Park (BINP), a known foci for spillover events.

### Methods

Focus groups (n = 24 groups) with 153 participants were convened and interview guide (See S1 Text) was used to conduct discussions in rural and urban settings surrounding BINP in Southwestern Uganda. Mixed methods were used for data analysis. For qualitative data, thematic analysis was used to identify and organize patterns of meaning related to the study objectives. Inductive coding and deductive analysis using a codebook was used to explore key themes associated with community understanding of zoonotic infections and participation in high-risk activities. For the quantitative analysis, a count data set was generated using the themes, sub-themes,

**Data availability statement:** We made all the relevant data available as summarized supporting information files. Interview data will not be shared because of sensitive participant information they may contain. Any other questions about the study can be obtained by reaching out to the Research Coordinator at Bwindi Community Hospital (email: researchbwindihospital@gmail.com).

**Funding:** This work was supported by the National Institute of Allergy and Infectious Diseases (NIAID) Award # U01AI151814 to CKJ and TSE and National Institutes of Health Fogarty International Center Award #R21TW012608 to TSE, SK and MW. The funders had no role in study design, data collection and analysis, decision to publish, or preparation of the manuscript.

**Competing interests:** The authors have declared that no competing interests exist.

and codes from the codebook. Multivariable logistic regression was used to assess the association of demographic, geographic and occupational factors with zoonotic understanding.

## Results

Few participants believed animals could transmit diseases to humans (or vice versa), with rural, younger residents as well as those working in healthcare showing higher awareness. Interviews corroborated this finding noting that even when recognized as potentially unsafe, meat from diseased animals was rarely reported and often consumed or resold.

## Conclusions

Misconceptions about zoonotic disease transmission are prevalent in Southwestern Uganda despite high-risk for spillover. Policy makers and government entities should prioritize culturally appropriate community education, contextually relevant mitigation of potential spillover events, and comprehensive research into drivers of high-risk activities.

### Author summary

Emerging and re-emerging infectious zoonotic diseases continue to pose significant threats to human and animal health. In Uganda, there is a growing encroachment into forests resulting in increased contact with wild animals and high-risk for spillover events. Despite the long history of zoonotic outbreaks in Uganda, there is a low awareness of zoonotic diseases. We conducted this study to understand communities' knowledge of the potential disease risks associated with contact with wild and domestic animals around the Bwindi Impenetrable Forest. Our research highlights economic drivers of unsafe meat consumption along with low awareness of zoonotic spillover risk. We described practices in rural and urban communities in Southwestern Uganda that increase the risk of zoonotic disease outbreaks. We found that a common but risky practice was the consuming of meat from sick or dead animals. This was especially apparent in low socio-economic status communities. Surprisingly, rural communities demonstrated a higher understanding of zoonotic disease transmission compared to urban populations. We noted that there was limited knowledge of the potential for zoonotic disease spread. Our findings emphasized the need for enhanced education efforts regarding zoonotic disease transmission and prevention. We recommend that rural communities, which are engaged in high-risk activities, receive education regarding eliminating practices that increase the risk of zoonotic infection. Additional research is also needed to further elucidate the drivers of misunderstandings about zoonoses and potential impact of educational or other interventions, given complex socio-economic conflicts with meat consumption despite known disease status.

## Introduction

The world is experiencing a period of accelerated emergence of pathogens with pandemic potential over the last century [1]. Zoonoses present one of the greatest threats to human health in the 21st century representing close to three-quarters of all newly emerging and re-emerging diseases [2,3]. In low-income settings, zoonoses are estimated to contribute to a quarter of the disability-adjusted life years (DALYs) lost when compared to 1% of DALYs lost in high income countries [4].

Uganda is known to host a wide range of pathogens of zoonotic origin. Some of these pathogens were first identified in Uganda including: West Nile virus (1937), Bwamba virus (1937), Semliki Forest virus (1942), Bunyamwera virus (1943), Zika virus (1947), O'Nyong'nyong virus (1959), and Bundibugyo ebolavirus (2007) [5,6]. In the last decade, Uganda has experienced outbreaks of swine fever (2013), Marburg disease (2016), yellow fever (2016), Crimean-Congo hemorrhagic fever (2018–2019), and Rift Valley fever (2018), as well as Ebola disease (ED) outbreaks in 2000, 2014, 2017, 2018 and in 2022 [7–13]. A seminal study in 2018 found significant serological evidence of exposure to ebolaviruses in the Bwindi region despite having no confirmed cases of Ebola disease [14].

Uganda remains a hotspot for pandemic threats due to its rich biodiversity, wide range of wild lands, rapid population growth, and increasing proximity to wildlife reservoirs [15,16]. The Uganda National Housing census report of 2024 shows that, it has a population of 45.9 million people, with a significant proportion dependent on agriculture such as livestock production [17]. According to the National Livestock Census of 2021, the country is estimated to have 2.27 million, cattle, 3.30 million goats, 0.74 million sheep, 2.22 million pigs, and 5.5 million poultry [18]. Endemic and re-emerging zoonotic diseases such as Anthrax and Rift Valley Fever in Uganda have had significant socioeconomic impacts, including morbidity and mortality resulting in the loss of productivity and income for livestock-dependent populations [19,20]. Accordingly, Uganda has adopted a One Health strategy aimed at addressing the complex interplay between human, animal, and environmental health [21]. With a vision of halving some key drivers of zoonotic infections, this strategy is intimately dependent on improving capacity to recognize threats, reduce high-risk contacts, and effectively respond to outbreaks. Rural communities are particularly important for this strategy given their disproportionate contact with both wild and domestic animals.

The Bwindi Impenetrable National Park (BINP) is a protected tropical rainforest in the Southwestern corner of Uganda with complex and biologically rich fauna and flora [22]. Forests in Southwestern Uganda are home to a number of wild animals with the potential to share zoonotic pathogens with humans due to proximity and shared resources [23]. The diverse fauna includes species with potential for harboring zoonotic diseases such as mountain gorillas (*Gorilla beringei beringei*), black-and-white colobus monkeys (*Colobus guereza occidentalis*), chimpanzee (*Pan troglodytes),* vervet monkeys (*Chlorocebus aethiops)*, red-tailed monkeys (*Cercopithecus ascanius schmidti*), olive baboons (*Papio anubis*), giant forest hogs (*Hylochoerus meinertzhageni*), cane rats (*Thryonomyidae spp*), bush pigs (*Potamochoerus larvatus*), black-fronted duikers (*Cephalophus nigrifrons*), and yellow-backed duiker (*Cephalophus silvicultor*). There are over 90 known bat (*Chiropteran)* species in Uganda, but new species continue to be identified, several of which are known to be reservoirs of zoonotic illness [24–26]. Ecotourism, including gorilla trekking, is a major attraction in the region, bringing in over 36,000 visitors to the forest annually [27]. These tourists may not always adhere to the guidelines preventing disease transmission between non-human primates and people [28].

The Southwestern region of Uganda supports one of the most rapidly growing populations in the country. The average reproductive rate per woman is 6 children although, among the poorest members of the population, it approaches 7.1 [29]. The combination of an expanding population and limited housing and agricultural land has resulted in encroachment into forests that have been undisturbed for centuries [30]. Illegal poaching of animals from local forests is not uncommon [31].

Despite the ever-increasing risk of zoonotic disease, there are studies conducted in other regions of Uganda that demonstrate minimal knowledge about zoonotic diseases and disease transmission [23]. In Kagadi district, livestock value chain actors had knowledge about zoonotic disease risk whereas their non-technical counterparts knew relatively little [32]. In several districts in Uganda, knowledge of zoonotic diseases that can be transmitted from humans to animals

(zooanthroponosis) is limited to diseases where there has been deliberate sensitization because of prior outbreaks or conservation efforts to limit transmission [32,33]. Using a mixed methods approach, the goal of this study was to better understand communities' contacts with wild and domestic animals and their knowledge of potential disease risks associated with these interactions.

## Methodology

### Ethics statement

Ethics clearance was obtained through UC Davis Institutional Review Board (#1614696–2), the Mbarara University of Science and Technology Institutional Review Board (#07/02–21) and the Uganda National Council of Science and Technology (UNCST) number HS1772ES. All participations were voluntary. Individuals were read an informed consent script and consent were documented by thumbprint or signature. For children, assent to participate in the study was obtained from parents followed by consent from the child was obtained. Written informed consent was obtained from every participant.

### Mixed methods approach

Given the complex and under-researched nature of zoonotic disease spread in these communities, policy makers and planners require a definable and culturally relevant understanding of key factors that contribute to outbreaks. The mixed methods approach utilized in this study is aimed at exploring factors involved in zoonotic disease spread and then beginning the process of quantifying the extent of these drivers within the local community. Combined, this data can help develop a richer understanding of the local context alongside the breadth of potential opportunities for future research, education, and/or intervention.

### Study area and context

From November 2021 to August 2022, we studied rural and urban settings in Southwestern Uganda in the Kigezi region. Based on the 2024 housing census, more than 1.7 million people reside in this region [17]. Kabale was chosen for the urban setting as it is a region of high human density and the closest urban center to the BINP. Three districts including Kisoro, Kanungu, and Rubanda districts were chosen for the rural setting because they border the BINP, a prominent ecotourism site. The urban and rural sites were approximately 50 miles apart, allowing for strong ecological comparisons. BINP is a UNESCO World Heritage Site that is home to a very biodiverse fauna including several species of non-human primates (NHPs) bats, birds, and small mammals [24]. Ecotourism, encroaching of farming practices, and illegal hunting create multiple avenues for contact with wild animals and risk for zoonotic spillover.

### Participants and sampling strategy

Snowball sampling utilizing local networks was selected as it was anticipated to be the most effective method of gathering potential participants in the targeted groups and was consistent with local cultural practices. Accordingly, community members participating in the focus group discussions (FGD) were recruited through local administrators, Community Health Workers (CHWs), and elders. Participants who lived within the selected study sites (Kabale, Kisoro, Kanungu, and Rubanda, see Fig 1), and who were 18 years old or greater and had resided in the community for at least five years were assumed to have adequate knowledge about their community and therefore eligible to participate. Individuals with direct or indirect wildlife contact were preferentially targeted for participation in the study. FGDs were conducted as only male, only female, or mixed gender (in the case of CHWs and Health workers). Each group was composed of six to seven participants. The interviews lasted between 60–120 minutes. Interviews were conducted in a quiet setting that was identified by the FGD members in each community.

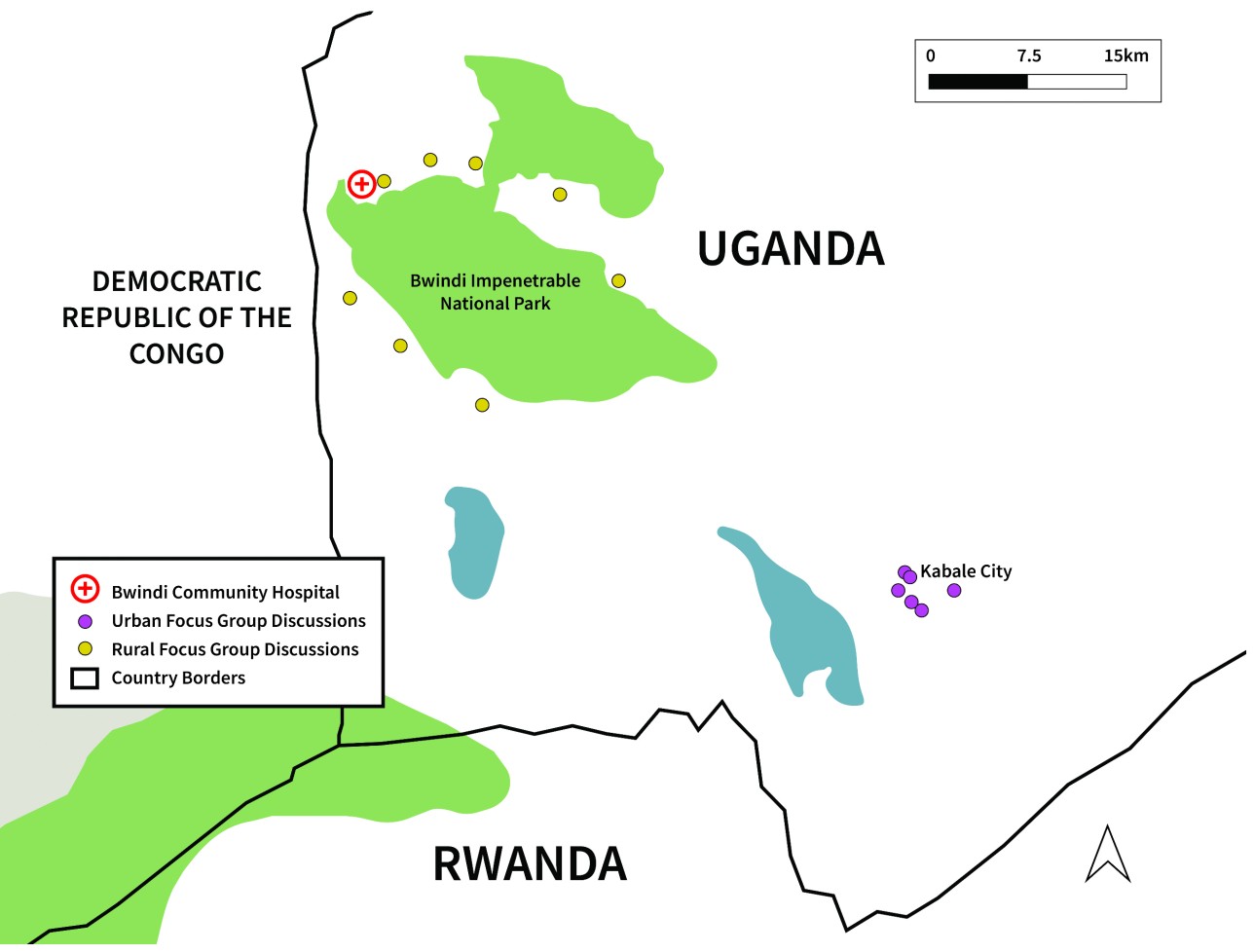

**Fig 1. A map showing locations of focus group discussions.**

## Data collection

FGDs were conducted as only male, only female, or mixed gender (in the case of CHWs and Health workers). Each group was composed of six to seven participants. The interviews lasted between 60–120 minutes. Interviews were conducted in a quiet setting that was identified by the FGD members in each community. A total of 24 focus groups with 153 participants from rural and urban communities were conducted (Table 1).

Socio-demographic data collected from each participant included age, gender, occupation, level of education and whether participant was a resident of a rural or urban community. Education was categorized into three levels: (i) Primary education, which encompasses the first seven years of schooling; (ii) Secondary education, which follows primary education and spans six years, divided into two stages Ordinary level (four years) and Advanced level (two years); and (iii) Tertiary education, which begins after the completion of secondary education. No participants withdrew during the course of the study. Final study size was driven by data saturation around key study themes.

The interviews were conducted by trained research assistants who were native speakers of Rukiga and Rufumbira (HN, MC, AI, PK, and NF). The study team was composed of several local and international collaborators with experience in qualitative research. The local collaborators (NH, BM, GM, BS, CG, CA, NF) were native to the community and fluent

**Table 1. Socio-demographic characteristics of focus group participants (n = 153).**

| Characteristic | Rubanda (%) | Kabale (%) | Kanungu (%) | Kisoro (%) |
|---|---|---|---|---|
| Sex | | | | |
| Male | 9/18 (50) | 30/63 (47.6) | 20/44 (45.5) | 10/28 (35.7) |
| Female | 9/18 (50) | 33/63 (52.4) | 24/44 (54.5) | 18/28 (64.3) |
| Tribal Group | | | | |
| Bakiga | 18/18 (100.0) | 63/63 (100.0) | 44/44 (100) | 22/28 (78.6) |
| Bafumbira | 0/18 (0.0) | 0/63 (0.0) | 0/44 (0) | 5/28 (17.8) |
| Other | 0/18 (0.0) | 0/63 (0.0) | 0/44 (0) | 1/28 (3.6) |
| Age in Years | | | | |
| < 35 years | 5/18 (27.8) | 19/63 (30.2) | 4/44(31.8) | 10/28 (35.7) |
| > 35 years | 13/18 (72.2) | 44/63 (69.8) | 30/44 (68.2) | 18/28 (64.3) |
| Occupation | | | | |
| Farming (Livestock, Mixed, Agriculture) | 16/18 (88.9) | 19/63 (30.1) | 30/44 (68.1) | 14/28 (50.0) |
| Health Worker | 0/18 (0.0) | 1/63 (1.6) | 8/44 (18.2) | 1/28 (3.6) |
| Other (Not involving contact with animals) | 2/18 (11.1) | 43/63 (68.2) | 16/44 (13.6) | 0/28 (0) |
| Education Level | | | | |
| None | 0/18 (0.0) | 0/63 (0.0) | 5/44 (11.4) | 0/28 (0.0) |
| Primary | 9/18 (50.0) | 12/63 (19.0) | 10/44 (22.7) | 9/28 (32.1) |
| Secondary | 3/18 (16.7) | 14/63 (22.2) | 3/44 (6.8) | 9/28 (32.1) |
| Tertiary | 0/18 (0) | 10/63 (15.9) | 6/44 (13.6) | 5/28 (17.9) |
| Not Recorded | 6/18 (33.3) | 27/63 (42.9) | 20/44 (45.5) | 0/28 (0.0) |

in the local language. Several collaborators, were largely based outside of Uganda but had familiarity with the setting (TS, CKJ, JJ, ER, MW, NG), had been living for several years in the local community (JL) or had lived locally for several decades and spoke Rukiga fluently (SK).

## Data analysis

**Qualitative data analysis.** Audio recordings were transcribed directly from the local language to English by an independent translator who had no involvement in the data collection process. All transcribed text was analyzed using thematic analysis independently by four researchers (HN, JJ, JL and SK) using NVivo 12 software (Nvivo, Inc). First, these researchers to developed codes and concepts within the text based on topics discussed by participants. Next, they tied together conceptually related codes to generate themes and sub-themes such as medical treatments or disease symptoms. Finally, these were tied together into the final concepts related to topics such as the perceptions and local understanding of the meanings of zoonoses. Data analysts used an iterative approach involving written, verbal, and other analytic techniques methods to refine each piece of the analysis in arriving at the final themes, sub-themes, and codes used in the codebook and subsequent outcomes.

**Quantitative data analysis.** A data set used for quantitative analysis was generated using the themes, sub-themes, and codes from the codebook. We counted the number of times pre-defined subthemes were mentioned across all interviews. A frequency table showing the prevalence of each subtheme across the dataset was generated. Scoring and ranking data obtained from the focus group discussions were analyzed using descriptive and non-parametric statistical methods. Associations between zoonotic and anthropozoonotic understanding and demographic and geographic variables were initially evaluated by the 2-sided Fisher exact test or the $\chi^2$ test as appropriate, and associations were measured by odds ratios (ORs). Multivariable logistic regression was then used to assess the association of factors with zoonotic understanding for variables that were significant on bivariate analysis ($P < 0.1$). Variables were included if they significantly

improved model fit, based on the likelihood ratio test (*P*<0.1), while minimizing the Akaike information criterion (AIC). Overall model fit was assessed using the Hosmer-Lemeshow goodness-of-fit test.

## Results

### Socio-demographic characteristics of study participants

A total of 153 participants joined the discussions. By occupation, 50.3% of participants were farmers, 68.6% were aged 35 years and above while31.4% were aged under-35 years. 96.1% of the participants were from the Bakiga tribe the dominant tribe in southwestern Uganda, 54.9% were female while 45.1% were male. In this region, crop growing at subsistence level is practiced by over 95% of the households and mixed with some livestock (chickens, goats, sheep, cattle). Large scale livestock farming is uncommon in the region because people live on small pieces of land due to high population density of the region. Forty percent of the participants had attended primary levels of education (Table 1).

### Perception of animal to human disease spread

153 participants from multiple communities, age groups, occupations, and education levels participated in the study (Table 1) and contributed to the overall study themes and sub-themes (Table 4). There was generally a lack of awareness regarding animal to human disease transmission (Table 2). Only 61 of 153 (33%) participants were aware that animals could transmit diseases to humans. People from rural settings were 3.53 times more likely to be aware that animals can transmit diseases to humans (OR = 3.53, p=0.017, CI: 1.19 – 12.65). The most commonly reported perceived mode of transmission of zoonotic pathogens was through animal products (24.5%), followed by contact with animals such as cane rats, mice, baboons among others (16.4%), and domestic animals (11.5%) (Table 2). The most mentioned diseases, known to be transmitted zoonotically were brucellosis, malaria, plague, rabies, jiggers, diarrhea, and Ebola. Participants believed that there was an increased risk of transmission of diseases where there was sharing of resources such as food, a water source or living environment. In most of these rural communities, animals are raised on free range and it's not uncommon for domesticated animals such as pigs to share food or contaminate food meant to be consumed by animals as they scavenge for food on their own. At the same time, domesticated animals are rarely seen as a source of infections to humans since people have lived with them for many generations and thus their perceived risk of zoonotic transmission is low.

> *"Diseases that animals transmit to people, is where you find someone eating food which a pig had also eaten, there you suffer from diarrhea" (FGD Men, Ruhija).*

Rats were thought to host multiple diseases (n=3 mentions) that could be transmitted to humans and bats (n=5 mentions) were alleged to deposit diseases on whatever surfaces they encountered. For the forested communities, rats constantly move in between the forest and the homes at the forest edge. In both urban and rural communities however, the perceived risk of disease transmission from rats to humans was low. For example, food could not be thrown away just because it had chew marks from rats. Instead, what would be done was that a piece that had chew marks would be cut off and the food (such as sweet potatoes, cassava, and fruits like bananas) would be consumed without additional measures to disinfect the food. We did not get a mention of instances where food would be completely thrown away in the discussions conducted.

Further, meat is highly prized and limited, particularly since hunting is prohibited in the BINP. We assessed how community members dealt with products from apparently sick (i.e., presenting with obvious clinical signs of a known or unknown infection such as irregular eating habits, sudden weight loss, vomiting, unusual stool, and lethargy) domesticated animals. These animals included domestic livestock, such as goats, cows, swine, and poultry. Most participants (n=34 mentions) in both rural and urban settings reported even if they knew that the meat was from a diseased animal it would be eaten.

**Table 2. Factors associated with awareness of zoonotic transmission in rural and urban populations.**

| Characteristic | Rural population Proportion (%) | Urban population Proportion (%) | OR (95% CI) | P-value |
|---|---|---|---|---|
| Overall zoonotic awareness | 21/90 (23.3) | 5/63 (7.9) | 3.53 (1.19 – 12.65) | 0.01* |
| *Awareness of specific zoonotic transmission mechanisms* | | | | |
| Consumption of animal products | 6/90 (6.7) | 9/63 (14.3) | 0.43 (0.12 – 1.45) | 0.12 |
| Contact with animal vermin | 6/90 (6.7) | 4/63 (6.3) | 1.05 (0.24 – 5.30) | 0.94 |
| Contact with domestic animals | 2/90 (2.2) | 5/63 (7.9) | 0.26 (0.03 – 1.69) | 0.10 |
| Other transmission mechanisms | 13/90 (14.4) | 9/63 (14.3) | 1.01 (0.37 – 2.89) | 0.98 |
| *Awareness of specific zoonotic diseases transmitted* | | | | |
| Brucella | 6/90 (6.7) | 4/63 (6.3) | 1.68 (0.36 – 8.30) | 0.50 |
| Respiratory diseases | 5/90 (5.6) | 9/63 (14.3) | 0.35 (0.09 – 1.26) | 0.06 |
| Ectoparasites (Flea Infestation and jiggers) | 2/90 (2.2) | 4/63 (6.3) | 0.34 (0.03 – 2.44) | 0.20 |
| Hemorrhagic diseases | 4/90 (4.4) | 1/63 (1.6) | 2.79 (0.27 – 139.38) | 0.35 |
| Diarrheal diseases | 5/90 (5.6) | 0/63 (0.0) | – | – |

*Significant association for p<0.05*.

Some participants reported that a sick animal would either be treated by the participant using local herbs or by simply buying medicine from the veterinary shops without a prescription or taken to a veterinary practitioner. If there was no hope of recovery, the animal would be sold quickly to a butcher at a reduced price before the clinical signs became obvious. There were reports where veterinary doctors and local leaders were bribed to allow meat from diseased animals to be sold.

*"They bribe (the Local Council) chairmen and the veterinary doctor so they can slaughter a sick animal to change the narrative; a cow which died because it was sick is pronounced to have been killed by a rope", (Men, Kabale City).*

In some instances, meat from diseased animals killed would be sold alongside or even mixed in with meats from freshly killed healthy animals.

*"Sometimes they slaughter the meat of the sick one and mix it with the healthy one", (VHTs Mpungu).*

Other reasons given for the consumption of potentially diseased meat included extreme hunger due to poverty and potential economic loss.

*"People eat the meat because of poverty",* (*VHTs Southern division*).

Even when the owner was willing to bury, there were those who would still ask for dead animals to consume them …. *"A friend of ours lost very many chickens and when someone found him burying them, he asked him if he can take one. With caution, he gave one to him and told him not to mention they came from him; because he wouldn't have personally given away a dead chicken", (FGD Men, Southern sector).*
*Illustration by Eunah Preston, UC Davis, One Health Institute and used with permission. This figure is published under the CC BY 4.0 license.*
Fig 2 summarizes community knowledge about transmission from most mentioned to least mentioned (left to right). The following specific transmission pathways were described by at least one FGD: brucellosis was known to be transmitted through the consumption of animal products such as meat and milk; Ebola through contact with

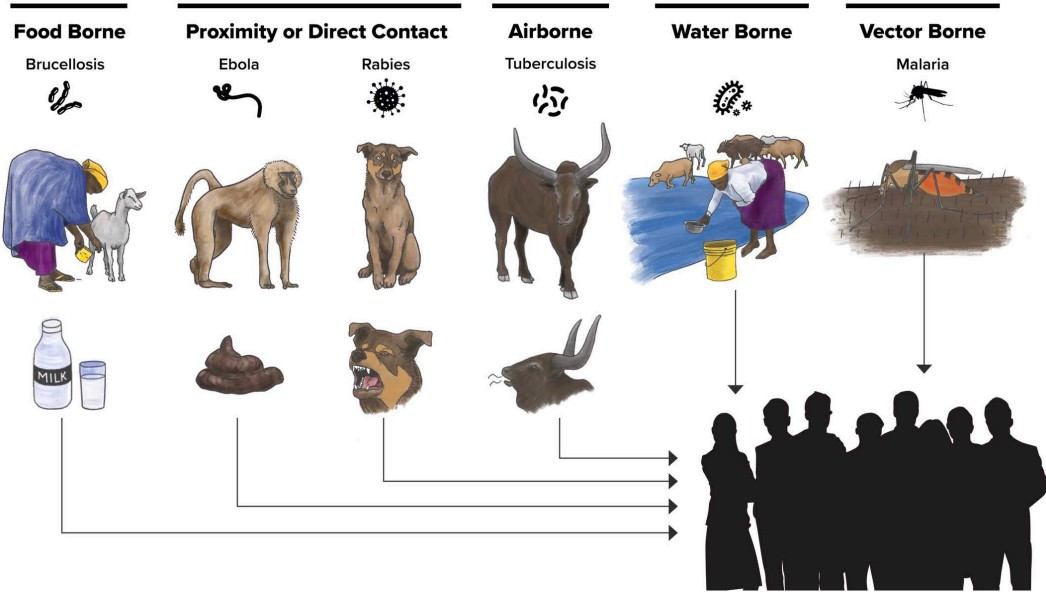

**Fig 2. Community knowledge about transmission of specific zoonotic diseases.**

primate bodily fluids; rabies through the bite of an infected dog; tuberculosis through breathing the same air as an infected cow; diarrheal diseases through contaminated water; malaria through the bite of a mosquito. Mosquitoes were largely not perceived to transmit zoonotic illnesses although many participants were aware that they transmitted malaria.

Results from our multivariable model (see Table 3) showed that participants from rural areas were significantly more likely to be aware of zoonotic disease transmission compared to participants who were from urban areas (OR = 3.98; P = 0.01). Participants who were greater than 35 years of age were also significantly less likely to have zoonotic disease awareness compared to those who were younger (OR = 0.4; P = 0.02). Participants who were trained as health-care workers (CHWs and professional healthcare givers) were significantly more likely to have zoonotic disease awareness (OR = 7.24; P = 0.02) compared to other occupations.

## High-risk contacts with animals

Participants mentioned that there was significant human contact with both wild and domestic animals in rural and urban locations (Table 4) even as participants from these groups varied in their awareness of these contacts (Table 3). Most wildlife contact occurred as participants were attempting to eliminate marauding animals (*n = 16 mentions*) in both domestic settings and in the farming areas at the forest edge. Wild animals (e.g., rodents, bats and non-human primates) were eliminated via poisoning of trapping (using rodent traps and cages for non-human primates) and then killing. Carcasses were handled with bare hands in most cases and in some instances (*n = 15 mentions*), polythene/pieces of paper (*n = 5 mentions*), leaves/stick (*n = 6 mentions*) or farm tools (*n = 2 mentions*) were used to avoid direct contact in the case of rodents and bats. There is recognition that disposal of animal remains requires contact with animals that can expose people to diseases.

*"Another way people get in contact with animals is when we build traps…. once the animal falls in the trap, they can't leave it there, it is the responsibility of the people to remove it so that it doesn't stay and smell or cause other troubles, they decide to dispose of it"* (FGD Women, Karangara).

**Neglected Tropical Diseases**

**Table 3. Demographic factors associated with overall zoonotic disease awareness based on final multivariable analysis.**

| Demographic factor | Proportion aware in group with demographic factor (%) | Proportion aware in group without demographic factor (%) | OR (95% CI) | *P-Value* |
|---|---|---|---|---|
| Rural | 21/90 (23.3) | 5/63 (7.9) | 3.98 (1.38 – 11.47) | 0.01 |
| Age (>35) | 34/105 (32.4) | 27/48 (56.3) | 0.40 (0.19 – 0.84) | 0.02 |
| Healthcare worker | 8/10 (80.0) | 53/143 (37.1) | 7.24 (1.41-37.27) | 0.02 |

*Significant association for p<0.05*.

**Table 4. Emerging themes and sub themes from focus group discussions.**

| Theme | Subtheme |
|---|---|
| Perceptions of animal to human disease spread | Limited awareness of animal to human disease spread |
| | Animal products, contact with animals (particularly vermin), or sharing food with animals were acknowledged vectors |
| | Differences were seen based on location (urban vs. rural), age (over or under 35), and occupation (health worker vs. other) in their perception of transmission risk |
| | Even when noted to have signs of illness, both domestic and wild animals were usually consumed |
| | Market for diseased meat enabled local spread |
| | High poverty levels meant animal illness did not prevent sale or consumption of meat |
| High-risk contacts with animals | Trapping and control of wild animals were common contact points |
| | Protective equipment was rarely used with living or deceased animals |
| | Domestic and wild animals often entered dwellings |
| | Encroachment of forest animals led to unintentional contacts |
| | High poverty levels meant animal illness did not prevent sale or consumption of meat |
| Human to animal spread | Poor human sanitation enabled spread |
| | Mountain gorillas and monkeys are particularly vulnerable |
| | Respiratory and diarrheal disease most important |
| Reporting disease outbreaks | Limited knowledge of reporting mechanisms |
| | Willingness to report was related to perceived disease severity and likelihood of receiving support |

Although these animals are known to harbor a wide range of pathogens, infection control and prevention were minimally observed while handling carcasses.

*"We use our hands to unset the trap and to pick the dead animals"* (FGD Women, Butobere).

*"Some people even carry dead corpses of some animals on their backs", (FGD Men, Katojo)*

Only in a few instances was infection control considered. The main methods used to clean rodent traps were washing with soap *(n = 2 mentions)* or pouring hot ash on the traps *(n = 1 mention).* Unintentional contact *(n = 15 mentions)* occurred when both domestic and wild animals, such as cats, dogs, and snakes, entered dwelling places. Occasionally domestic animals entered the beds of sleeping occupants or ate some of their food which was left uncovered and contaminated it.

*"For cats, sometimes you find it sleeping with you in the blanket, then you touch it"* (FGD Female, Kabale City).

Unintentional contact also occurred at the forest edges.

*"Mostly we that neighbor the park, we have the monkeys which are very common and usually come to where we graze our goats, so sometimes these monkeys leave their hairs with our goats when they get in contact with them, and we definitely get into contact with their hairs" (FGD Male, Nyamishamba).*

**Human to animal disease spread**

Participants were also minimally aware of the human-to-animal transmission of diseases (n = 26/153), representing only 17% of all study participants. Diseases that were reported to transmit from humans to animals, in descending order, included respiratory, diarrhea, helminths, Ebola, fever/skin rash, and measles. The most commonly mentioned mode of transmission was open defecation.

*"If you defecate in an open space and an animal eats the fecal material, it can get sick …." (FGD Women, Nkuringo).*

Animals that were believed to be at the highest risk of human-to-animal transmission were mountain gorillas and monkeys. Transmission knowledge was skewed toward transmission to gorillas (n = 12/26). Some participants in the focus group noted that mountain gorillas are at risk of contracting respiratory infections (e.g., influenza, cough, SARS-CoV-2) from infectious humans.

*"…. animals also get infections from us, because (the Uganda Wildlife Authority) taught us that also the gorilla can suffer from flu with a nasal discharge. You find it isolating itself from the members of the family, moving more slowly, and raising the hair on its skin in that way it is also sick and mostly infected from a human." (FGD, Ruhija, Men).*

Diarrheal diseases were also thought to be transmitted from people to non-human primates, such as mountain gorillas, monkeys, and chimpanzees, through inappropriate sewage disposal. Fecal matter was mentioned as a transmission mechanism through which intestinal parasites could be spread to non-human primates, especially if the human fecal matter was left unburied or improperly disposed. This was noted to occur around homes near the park edge or within the park as people transverse through the forest.

*"For example, those gorillas, when someone defecates near the road, it will come touching everywhere, then ends up eating it in the grass, resulting in it suffering from worms, that is another way they get infections from people." (FGD Men, Ruhija).*

**Reporting disease outbreaks**

We assessed participants' willingness to report diseases in humans and animals. A small proportion of the respondents (n = 28/153) were aware of where to report disease outbreaks. When inclined, respondents reported a disease outbreak to a CHW (n = 12/28), a local health facility (n = 8/28), and, in some instances, to local leaders (n = 5/28).

The choice of where to report was predicated on the assistance that they hoped to receive, the venue most available to them, and the perceived infectiousness of the disease.

*"In case there was an outbreak of such a disease, they first inform the Village Health Teams (VHT's). The VHT's have numbers of hospitals and responsible people and look for a car" (FGD, Women Karangara).*

Common reasons for not reporting were not knowing where and whom to contact.

*"It is unlikely for a rural person to report any case because first of all, they don't know there is someone they can report to, instead of the health care system they go to the local government system, to local leaders, or call the police…..." (FGD Health Workers, Bwindi).*

## Discussion

Emerging infectious diseases with pandemic potential threaten both human and animal health. Despite Uganda's frequency of outbreaks of zoonotic diseases, rich biodiversity in conserved parks, rapid population growth and human encroachment into forests (1), the level of awareness amongst participants in this study was low.

Our research augments previous studies into communities' views of zoonotic diseases in East Africa demonstrating a general lack of awareness of zoonotic disease transmission. We found that only 1/3 of the respondents were aware of zoonotic diseases. A Tanzanian study found differences in understanding of zoonotic diseases among pastoralists from different parts of the country with differing educational efforts [34]. A study among actors in the livestock trade in the Lake Victoria crescent ecosystem in East Africa found that although there was awareness of livestock diseases, there was a lack of awareness of which diseases could be transmitted from livestock to humans. There were also many misconceptions regarding the origins of zoonotic diseases and methods of disease spread [35]. A study in high-risk districts of Western Uganda for Ebola outbreaks found that only 30% of the respondents described knowing that Ebola could be transmitted from animals [36]. Other studies have also reported misconceptions about the source of zoonotic infections including the spiritual, airborne, and mosquito transmission of hemorrhagic fevers. Providing education in this regard would reap great rewards in local citizens' understanding of the diseases that commonly circulate [37].

Our finding that rural residents had a greater understanding of zoonotic disease spread, disagrees with research in other regions which indicated that people in rural areas were either equal or less likely to understand zoonotic risks from wildlife, despite there being more contact with wildlife compared to urban areas. A study in Eastern Ethiopia found that the associations between the respondents' residence and knowledge of zoonotic Anthrax, rabies, and brucellosis were not significantly different [38]. The difference in awareness between rural and urban communities in our study can be partially explained by the presence of sustained community education programs aimed at preventing disease transmission from humans to wildlife such as the endangered mountain gorilla in BINP. Comparatively, there are no conservation areas in the Kabale urban setting. Education programs have been ongoing in communities around BINP about the risks of transmitting human diseases to mountain gorillas after several studies reported that humans were the source of zoonotic diseases such as scabies and intestinal helminths in the mountain gorillas (*Gorilla beringei beringei*) [39–41]. Surprisingly, we noted that sick or dead animals were not handled in a sanitary manner and were frequently consumed. This practice increases the risk of zoonotic transmission and spillover. Meat from sick animals has been consumed for many generations in these traditionally agricultural-hunter-gatherer communities. With closure of the open hunting around the national parks, meat became very expensive and therefore left for those more affluent. At the same time, meat consumption is deep-rooted within the culture and part of the most desirable components of the diet of the communities in the two sites. Dead animals provide opportunity for very cheap meat for those who couldn't otherwise afford it. In other parts of Uganda, there have been reports of Anthrax among people knowingly consuming meat from a sick animal [42]. Good knowledge about anthrax among residents of a repeatedly-affected community did not translate to safe practices [42,43]. Consumption of meet from sick animals is widespread in Africa with similar findings noted in Kenya where 60% of the participants had consumed meat from a sick animal [44] and Zambia, where people were aware that consuming beef from an infected cow would transmit Anthrax, but would continue to consume the meat. [45]. Respondents in our study noted that even if they were aware an animal may have carried a disease, it would still often be consumed either in the household where it died or, after negotiations with brokers, sold at local markets alongside regular meat. These findings underscore the complexity of imparting effective behavior change amidst competing socio-economic drivers. People in our study communities

and many others throughout sub-Saharan Africa place greater value on meat consumption despite any perceived risks associated with contracting zoonotic infectious diseases.

It was interesting to note that although very few of the participants were aware of human-to-animal transmission of disease, the rural areas were more cognizant than urban populations. Further research is needed to investigate the drivers of this knowledge gap as it was beyond the scope of this study. In addition, most study participants were from the Bakiga tribe and no people of Batwa ethnicity chose to participate. This may have been due to existing segregations between the two ethnic groups and Batwa feeling less comfortable participating. Further research is needed to understand the differences in awareness of zoonoses between these two tribes especially given the Batwa's historic residence within the local forest. Both park personnel and the general public were more aware of human-to-mountain gorilla transmission of illnesses compared to other wildlife suggesting a potential educational impact of recent campaigns on local knowledge of zoonotic disease although further investigation is needed particularly to understand if this outreach has led to changes in key outcomes such as high-risk contacts or disease spread and whether these results are replicable in other key groups [46].

Both quantitative and qualitative results of this study offer clues into potential local drivers of zoonotic spread in the region. Considering local and regional One Health strategies, results from this study can inform Uganda's One Health strategy but also be incorporated into existing education efforts such as the World Organization for Animal Health's ongoing education programs for community animal health workers or other regional efforts such as ZOOSURSY [47]. The complex interplay between drivers of behaviors further supports the need in this community for the co-designed, contextually relevant interventions.

While providing a useful starting point for future study and interventions, this study had several limitations. One key limitation was that having used the Snowball sampling approach to recruit participants, there is a likelihood that participants might have identified those with similar characteristics, attitudes, or experiences. Similarly, while we attempted to quantify beliefs and attitudes amongst these important informants, the data collected only reflect those sampled within the context of the study limiting their generalizability to the wider population.

## Conclusions

This study provides a detailed analysis of residents near the Bwindi Impenetrable Forest regarding their perception of zoonotic spread of disease. Our findings indicate that only one-third of those surveyed were aware of the potential of animal to human transmission of disease. Surprisingly, those residing in rural areas were more knowledgeable than those in urban settings. Our study also indicated that participants rarely properly disposed of sick animals and that these animals were commonly consumed. Importantly, it was unlikely that dead or diseased animals were properly reported to appropriate authorities. Based on our findings, policy makers and government entities should prioritize education regarding disease spread between humans and animals, prevention of known potential spillover events like consumption of diseased meat or contact with wild animals, and research into the drivers, including poverty and wildlife encroachment, of these high-risk activities.

While providing knowledge and guidance is essential, without an approach that considers context, competing motivations, and structural/cultural barriers, behavior change is difficult to attain. Although on a purely intellectual level, providing vital information seemingly would create opportunities for improvement; however, sustainable behavioral change requires a more integrated approach. This approach needs to consider the complexities of deeply ingrained patterns. Human behavior is complex, particularly when the cause of dysfunction is abject poverty. Behavior is shaped by context. It is abundantly apparent that social, cultural and particularly economic factors take precedence over health goals. Consequently, we have found the necessity of shifting our messaging to a participant-focused approach. As Ugandan as well as most of African culture is relationship based, without acknowledging critical relationships, positively impacting behavioral change is nearly impossible. One of the ways we propose is to expand the one health workforces in communities at the greatest risk of zoonotic spillover. In Uganda and most of Africa, lay Community Health Workers (also called Village

health teams (VHTs) in Uganda) play an important role in providing education to communities. Adding training about zoonoses to their curriculum would increase their ability to provide education to the communities within which they live to identify, report and respond to. Around the Bwindi Impenetrable National Park and Kabale city, there are over 1000 VHTs, each responsible for 20–25 households and play a role as eyes, ears and feet on the ground for community health care engagement. The training of VHTs can be expanded to include zoonotic aspects in their curriculum on disease prevention, reporting of animal and human outbreaks and dissemination of health information in a culturally sensitive manner. Around the Bwindi and Kabale communities the VHTs are valued purveyors of medical information for the isolated villages in our region giving us feedback and advice regarding the success of our interventions. It is our impression that this method of engagement and information sharing will engender a more robust behavioral change but more research is needed to further define the value of this approach.

Population increases and animal habitat degradation in Uganda and other parts of the world will continue to increase human contact with zoonotic infections, making zoonoses education campaigns and other culturally relevant interventions well worth the investment.

## Supporting information

**S1 Table.  Factors associated with awareness of zoonotic transmission in rural and urban populations.**
(XLS)

**S2 Table.  Demographic factors associated with overall zoonotic disease awareness based on final multivariable analysis.**
(XLSX)

**S1 Text.  English focus discussion guide.**
(DOCX)

## Acknowledgments

We also thank Bwindi Community Hospital, Gorilla Doctors (Mountain Gorilla Veterinary Project, Inc.), Rugarama Hospital, Uganda Wildlife Authority, and the District Health Office of Kabale and Kanungu for their support towards the research project.

## Author contributions

**Conceptualization:** Birungi Mutahunga, Scott Kellermann, Christine Kreuder Johnson, Tierra Smiley Evans.

**Data curation:** Scott Kellermann, Felista Nanono, Prossy Katushabe.

**Formal analysis:** Nahabwe Haven, Jalika Joyner, Julia Lippert, Gilbert Mateeka, Charlotte Aguti, Isaac Ahwera, Tierra Smiley Evans.

**Funding acquisition:** Micheal Wilkes, Christine Kreuder Johnson, Tierra Smiley Evans.

**Investigation:** Nahabwe Haven, Charles Mugisa, Felista Nanono, Prossy Katushabe.

**Methodology:** Nahabwe Haven, Christine Kreuder Johnson.

**Supervision:** Nahabwe Haven, Birungi Mutahunga.

**Validation:** Nahabwe Haven.

**Visualization:** Julia Lippert.

**Writing – original draft:** Nahabwe Haven, Scott Kellermann, Jalika Joyner, Nicole R Gardner, Tierra Smiley Evans.

**Writing – review & editing:** Nahabwe Haven, Birungi Mutahunga, Scott Kellermann, Jalika Joyner, Julia Lippert, Evan Andrew Rusoja, Micheal Wilkes, Gilbert Mateeka, Benard Ssebide, Charlotte Aguti, Nicole R Gardner, Tierra Smiley Evans.

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
