## [Decision Letter · Decision Letter 0]

5 Feb 2025

PNTD-D-24-01469

Assessment of exposure to zoonoses and perceptions of zoonotic transmission surrounding the Bwindi Impenetrable Forest, Uganda

Dear Dr. Haven,

Thank you for submitting your manuscript to PLOS Neglected Tropical Diseases. After careful consideration, we feel that it has merit but does not fully meet PLOS Neglected Tropical Diseases's publication criteria as it currently stands. Therefore, we invite you to submit a revised version of the manuscript that addresses the points raised during the review process.

Please submit your revised manuscript within 60 days Apr 06 2025 11:59PM. If you will need more time than this to complete your revisions, please reply to this message or contact the journal office at plosntds@plos.org. Please include the following items when submitting your revised manuscript:

We look forward to receiving your revised manuscript.

Kind regards,

Philip Mshelbwala

Academic Editor

Victoria Brookes

Section Editor

Shaden Kamhawi

co-Editor-in-Chief

Paul Brindley

co-Editor-in-Chief

**Journal Requirements:**

At this stage, the following Authors/Authors require contributions: Scott Kellermann, and Tierra S. Evans. Please ensure that the full contributions of each author are acknowledged in the "Add/Edit/Remove Authors" section of our submission form.

2) Tables should not be uploaded as individual files. Please remove these files and include the Tables in your manuscript file as editable, cell-based objects. For more information about how to format tables, see our guidelines:

https://journals.plos.org/plosntds/s/tables

3) Some material included in your submission may be copyrighted. According to PLOSu2019s copyright policy, authors who use figures or other material (e.g., graphics, clipart, maps) from another author or copyright holder must demonstrate or obtain permission to publish this material under the Creative Commons Attribution 4.0 International (CC BY 4.0) License used by PLOS journals. Please closely review the details of PLOSu2019s copyright requirements here: PLOS Licenses and Copyright. If you need to request permissions from a copyright holder, you may use PLOS's Copyright Content Permission form.

Potential Copyright Issues:

- Figure 2; Please confirm whether you drew the images / clip-art within the figure panels by hand. If you did not draw the images, please provide a link to the source of the images or icons and their license / terms of use; or written permission from the copyright holder to publish the images or icons under our CC BY 4.0 license. Alternatively, you may replace the images with open source alternatives. See these open source resources you may use to replace images / clip-art:

- Figure 1; Please provide a direct link to the base layer of the map (i.e., the country or region border shape) and ensure this is also included in the figure legend; and provide a link to the terms of use / license information for the base layer image or shapefile. We cannot publish proprietary or copyrighted maps (e.g. Google Maps, Mapquest) and the terms of use for your map base layer must be compatible with our CC BY 4.0 license.

4) We note that your Data Availability Statement is currently as follows: "Data has been made available". Please confirm at this time whether or not your submission contains all raw data required to replicate the results of your study. Authors must share the “minimal data set” for their submission. PLOS defines the minimal data set to consist of the data required to replicate all study findings reported in the article, as well as related metadata and methods (https://journals.plos.org/plosone/s/data-availability#loc-minimal-data-set-definition).

- The points extracted from images for analysis..

**Reviewers' Comments:**

Reviewer's Responses to Questions

**Key Review Criteria Required for Acceptance?**

**Methods:**

-Are the objectives of the study clearly articulated with a clear testable hypothesis stated?

-Is the study design appropriate to address the stated objectives?

-Is the population clearly described and appropriate for the hypothesis being tested?

-Is the sample size sufficient to ensure adequate power to address the hypothesis being tested?

-Were correct statistical analysis used to support conclusions?

-Are there concerns about ethical or regulatory requirements being met?

Reviewer #1: This is a mixed-methods study - but at least for the qualitative aspects, the authors could consider referring to the Standards for Reporting Qualitative Research (SRQR), available at http://www.equator-network.org/reporting-guidelines/srqr/. There are some aspects of the methodology which to me are unclear and would benefit from more structure. For example:

- Rather than a section titled “Focus group discussion”, I would expect rather to have sections titled “Data collection” where the method of data collection is provided, “Participants”, where information on the choice of participants and sampling strategy is provided, and “Researcher characteristics”, describing the characteristics of the researchers (their experience in qualitative research, language, prior relationship with participants (if at all), positioning in relation to participants (e.g. outsider or insider) – as this information has implications for the interpretation of the results. Please see the SRQR for further details.

- The abstract mentions a three-stage coding framework and the use of grounded theory – but there is no mention of this at all in the methodology section. In fact, aside from the abstract, there is no indication at all of what methodological approach was used to analyse the qualitative data. If grounded theory was used, this appears to me a bit counterintuitive, as the themes appear to have been defined almost in advance and in relation to the research topic of interest – this is a perfectly valid way of doing it, but does not really seem compatible with a grounded theory approach (which would involve the researchers’ identifying or generating the themes from the data itself). Some additional explanation and clarity on this would be appreciated. I would recommend including a paragraph on “Qualitative approach” where this could be described.

Reviewer #2: The study ethics was cleared through international university review board and the in-country review board and national Council of Science and Technology and participants were read an informed consent information and signed off. There was no mention of participants withdrawing from the study.

The goal of the study was to better understand communities (rural and urban) contacts with wild and domestic animals and their knowledge of potential disease risks associated with these interactions. The reviewer recommends the authors state in the aim/goal that the study was conducted near the Bwindi Impenetrable Forest in Uganda a known hotspot for zoonotic pathogen spillover. The review suggests awareness be added to the study goal given table 2 of the results is about awareness of animal associated transmission risk factors and specific zoonotic diseases.

In the study (total number of participants = 153) qualitative methods of focus group discussions using a questionnaire interview guide were used. In the Rural focus groups, there were 90 people split into groups of 6-7 people across three rural districts of Kisoro, Kanungu and Rubanda. The group size for each focus group was appropriate but referring to table 1 there appears to be only two livestock farmers one from urban group and one from rural Kanungu. Table 1 should describe if agriculture/farming have livestock, what type of livestock or are hunters (what they hunt)? More description about participants contact with animals or ownership of animals is required. Also were any participants involved as guides for ecotourism associated with non-human primates etc? The reviewer did not have access to supplementary material Appendix 1 therefore could not review the interview guide.

The urban FGD included 63 people (live in Kabale City) about 50 miles away from the rural districts. There were 30 percent fewer urban participants in FGD than in rural groups. The two comparison groups urban or rural should have approximately the same number of participants given the study is assessing the odds (probability) of a zoonosis or zooanthroponosis knowledge and transmission pathways between different geographic locations (rural and urban) and exposure to animals/humans or animal products.

More information is needed about how the community members were recruited through local administration, community health workers and elders. The only inclusion criteria mentioned were being 18 years old or greater in age and residing in the community for at least five years to have adaptive knowledge. The CHW may have been biased in their selection choosing participants who had contacted a zoonotic disease. Was the sampling convenience or purposive sampling?

The main tribal group in the study were the Bakiga people therefore it is not possible to compare between Bafumbira and Bakiga people. It is noted that the Bantu people are the original forest dwellers and were not included in the focus groups. This tribe were hunter and gatherers and were evicted from the BINP in 1991 when the park was formed. The authors should discuss why one tribe mainly participated in the study and how they could have engaged other tribes in the study to have a better representation of the communities in this area. The socio-economic situation of the communities could also be described given those bordering Bwindi are the poorest in the region and have limited household income alternatives (Bitariho et al Conservation Science and Practice. 2022;4:e12761).

Was the interview guide (questionnaire) pretested with a few urban and rural people (representative of survey participants) to check the questions were clear and not ambiguous and culturally/gender appropriate? https://www.fao.org/4/w3241e/w3241e05.htm? Pretesting (piloting) of interview guides is recommended and best practice for studies.

The statistical analysis in the mixed methods descriptive study were appropriate though the confidence intervals for the odds ratios are required. Having an independent translator (not involved in data collection) transcribe the audio recordings to English reduced information bias that may occur if study FGD interviewers undertook this transcription. Additionally, the reviewer acknowledges study efforts to remove interpretation bias by having three researchers involved in the creation of themes and sub-themes related to perceptions and local understanding of the meaning of zoonosis.

Multivariable logistic regression was used to measure associations of risk factors with zoonotic understanding for variables that significantly improved the model fit.

Reviewer #3: The methods overall are well-described and appropriate. A few minor comments:

- Lines 128-129: please state if informed consents of illiterate participants were witnessed by an independent witness. If not, please confirm that the consenting procedure was approved as such by the local and national Ugandan ethics committees.

- Lines 152-155: please provide more information regarding how participants were recruited and selected for the focus groups. Was there an attempt to ensure the groups were representative of the local population (e.g., in terms of age, socioeconomic status, education, etc.)? Were residents selected at random to participate?

- Line 155: please describe or list the inclusion/exclusion criteria

**Results** :

-Does the analysis presented match the analysis plan?

-Are the results clearly and completely presented?

-Are the figures (Tables, Images) of sufficient quality for clarity?

Reviewer #1: No major comments, please see my attached comments for more detailed suggestions.

Reviewer #2: The receiver recommends the inclusion of a table showing the main themes and sub-themes that were created from the focus group discussions transcribed text. The perceptions and local understanding of the meaning of zoonoses are provided as examples of the main themes. What was the local understanding of the meaning of zoonoses? Were the specific symptoms of a particular illness in humans associated with symptoms in animals (syndromic framework) or was it associated with case definition and test result? Were photos or drawings used by the interviewer to convey questions?

The figure 2 of community knowledge about transmission of specific zoonotic diseases using pictures to illustrate the animals, vectors and animal products is well presented. In line 240 the reviewer notes a correction is needed to state that “many participants being unaware that they transmitted malaria”.

The table 2 and table 3 need the Odds Ratio 95% confidence intervals to be included as it is important to determine if the OR confidence interval includes the null value OR =1.

Table 2 is described as “factors associated with awareness of animal-to-human pathogen transmission” . The reviewer suggests changing the description to awareness of risk factors for zoonotic transmission and specific diseases. How is overall zoonotic awareness defined and calculated? Given the demographics of the study participants with 68.2 percent of participants in urban Kabale not having an occupation with animal contact, it is not surprising that the rural FGD participants were 3.53 times more likely to be aware that animals can transmit diseases to humans compared to urban participants.

The mode of transmission comment from FGD men, Ruhija (line 221-222) relating to eating food which a pig had also eaten and then suffering from diarrhea would need further investigation and could possibility be associated with transmission pathway for cysticercosis given Taenia solium is prevalent in pigs in many regions of Uganda?

It is interesting to note that bird flu (avian influenza) was mentioned once line 275 (zooanthroponoses) though respiratory diseases were included in table 2 therefore flu could be part of this specific zoonotic disease transmitted category. Another priority zoonosis is anthrax which was only mentioned in the discussion (line 395-396) of the paper associated with transmission from consumption of beef from sick cows. The One Health zoonotic disease prioritization for multisectoral engagement in Uganda (2017) included anthrax, zoonotic influenza viruses, viral haemorrhagic fevers, brucellosis, trypanosomiasis, plague and rabies. The Uganda One Health strategic plan 2018 -2022 had a goal to reduce the burden of prioritised zoonoses and AMR by 50 percent. The authors should discuss how their study may support Uganda to reduce the burden of prioritized zoonoses especially in this hotspot area.

In table 3 the demographic factors of rural, age and healthcare worker, were associated with overall zoonotic disease awareness. This table needs to define the “age” factor. From the OR = 0.40 it could be concluded that older participants are less aware of zoonotic diseases but given the small sample size it is important to include the confidence interval for this odds ratio result. It is logical to make the assumption that health care workers would have a significant greater zoonotic disease awareness given their training than other occupations (agriculture, livestock, farmers). It would be useful to further examine the ten health care workers education background and training.

The results description in lines 264-266 about proportion of participants with awareness of anthroponotic (animal- to- human) transmission mostly residing in rural setting (n=21/26) should be moved to the previous paragraph and it would be useful to know the occupations of these participants.

The transmission knowledge of diseases from humans to animals were skewed toward transmission to gorillas (n=12/26). It is noted that one of the study team is Dr Benard Ssebide from the Gorilla Doctors and had been involved with the wildlife authority. It would be useful to review the interview guide to better understand the questions associated with zooanthroponosis knowledge and conduct further studies to evaluate the Uganda Wildlife Authority training for Park personnel and general public (noted in last paragraph of discussion). Did all 12 participants who knew about this risk transmission pathway from humans to gorilla’s undertake this training in the past and were most in the FGD men from Ruhija?

The study results about contact with animals is well presented and an important finding in this study. The sentence in line 298-299 requires changing to read more clearly and could be “There is recognition that disposal of animal remains requires contact with animals that can expose people to diseases”. It is important to note that although some participants knew the animals could be infected with pathogens, they did not carry out appropriate infection prevention and biosecurity practices while handling carcasses. It would be useful to know if higher risk of zoonotic contact with animals was associated with gender? Is it mainly women who unset the animal traps? Gender was not found to be a demographic risk factor with overall zoonotic disease awareness as not included in table 3 results.

The study results on contact with meat from sick and dead animals are important findings. The statement that “people eat the meat because of poverty” by village health teams of southern division demonstrates the need to understand the drivers of behaviour and addressing the root causes of risk taking. A One Health approach is needed to support community health through improving livestock and wildlife health. Training of Community animal health workers (CAHW’s) is needed and the World Organisation for Animal Health WOAH (formally OIE) has recently launched competency and curriculum guidelines for community animal health workers which should be referenced. https://www.woah.org/app/uploads/2024/09/woah-competency-and-curriculum-guidelines-for-cahws-071024.pdf. Additionally, the WOAH and African Union commenced a project called ZOOSURSY to further build One Health collaborations for disease surveillance of priority emerging and re-emerging zoonotic diseases with implementation of the WOAH Wildlife health framework. The reviewer recommends that outcomes from this small qualitative study may support further research in this zoonosis hotspot area and more widely.

In the reporting of disease outbreaks results the willingness of participants to report diseases in humans and animals was found to be low (n=28/153, 18%) with half of those reporting contacting the community health workers. As there was no mention of community animal health workers or veterinarians for reporting disease occurrence in animals – perhaps participants were more focused on disease outbreaks in humans?

Reviewer #3: Overall, the results are presented in a clear and comprehensive fashion. There are just a few comments:

- Line 218, "diarrheal" should be "diarrhea".

- Table 3: please list the age groups (i.e., <35 yrs, >35 yrs) in the table or table legend. Also, some cells are missing the percentages in parentheses.

- Line 275: "influenza" not "Flue".

- Line 291: the first sentence of this section ("Contact with animals") appears to be missing a word.

- Line 294: a closed parenthesis is missing after "primates".

**Conclusions** :

-Are the conclusions supported by the data presented?

-Are the limitations of analysis clearly described?

-Do the authors discuss how these data can be helpful to advance our understanding of the topic under study?

-Is public health relevance addressed?

Reviewer #1: My general feeling is that the authors have prioritised their quantitative results over the qualitative results – with the outcome being that the discussion and conclusion focus overly on a “lack of awareness” rather than the reasons for the lack of awareness, and the broader context around managing zoonotic disease risk in the community. This can be seen in the abstract, for example, where only the quantitative findings are mentioned. One of the benefits of using qualitative approaches alongside quantitative is that rather than just saying “awareness was low because XX% were not aware of X” – is that you can also start to think about why that might be the case, and what could be done to address this. The conclusion regarding lack of awareness is supported by the data, but I think there are additional interesting and worthwhile conclusions that could be drawn from the results (for example -the mention of poverty as a driver for activities which put individuals at risk of exposure to potentially infected meat). Situating “lack of awareness” as one of multiple challenges facing these communities would, I feel, give the results more weight and help in drawing some conclusions for how to advance research and action on zoonotic disease management.

My concern really relates to the fact that the only recommendation of this paper is for policy makers and government to prioritise education regarding zoonotic diseases – it would be ideal if the authors could identify other recommendations on the basis of their results and to put these in a broader context.

Additional reflections upon the limitations of the study would be appreciated.

Reviewer #2: In the discussion several statements are missing references to support them – for example line 370 “our research augments previous studies into communities’ views of zoonotic diseases”.

The authors state that misconceptions about the origins of zoonotic diseases and transmission can be overcome through providing education citing a cross-sectional study in DRC following the end of the 2018 Ebola outbreak. The conclusion in this study recommended co-developed culturally sensitive and inclusive community evidence-based programs which compliments early anthropological insights from the West African Ebola epidemic about engaging communities and building trust. (Wilkinson A, Parker M, Martineau F, Leach M. 2017 Engaging ‘communities’: anthropological insights from the West African Ebola epidemic. Phil. Trans. R. Soc. B 372: 20160305. http://dx.doi.org/10.1098/rstb.2016.0305).

Increasing community knowledge and awareness alone is unlikely to be sufficient to successfully change human behaviours that drive zoonotic disease risk as described in a scoping review by Anna Durrance-Bagale et al (Drivers of zoonotic disease risk in the Indian subcontinent: A scoping review. One Health 2021. 13:100310). Co-designing context specific interventions using interdisciplinary and participatory methods are needed using One Health approaches. The cited (#31) Mangesho et al 2017 study among pastoralists in northern and eastern Tanzania also supports context specific interventions using One Health approaches taking into consideration social, cultural and economic aspects of communities.

Reviewer #3: In the Discussion, there are a few items that should be addressed:

- Lines 370-371: the sentence beginning, "While we found that only ⅓..." is a fragment.

- Lines 371-373:: pastoralists were compared to what other group(s)? Please clarify.

**Editorial and Data Presentation Modifications?**

Reviewer #1: (No Response)

Reviewer #2: Major revision required. Refer to comments in review.

Reviewer #3: (No Response)

**Summary and General Comments** :

Reviewer #1: This is an interesting paper on an important topic, and the importance and the context of this topic was well established in the introduction. The strengths of this paper include the focus on a topic of high public health importance and inclusion of a community of direct relevance for the topic, and the broad range of topics included for discussion with the participants. This is already a good paper, but I would like to recommend some few changes which I think could help to improve this paper further. My requests for revision relate principally to the methodology, and the discussion/conclusion.

- I think the methodology could benefit from some additional structure and detail, and my comments on the method I have recommended use of the Standards for Reporting Qualitative Research (SRQR). My main concern relates to the mention of grounded theory in the abstract, which is not mentioned in the methodology – nor is any other approach to the qualitative data. Additional information on this would be needed. Additionally, somewhere in the paper (either in the introduction or the discussion) to understand the rationale behind why a mixed-methods approach was chosen.

- I think the discussion could go further in putting the results in context and in the depth of analysis. For example, when observing that this study and many others have identified a lack of awareness – it may be useful to see if there are similarities which may contribute to the lack of awareness, or to consider the body of research that has suggested that awareness alone is not sufficient to change behaviour. In addition to mentioning papers that have similar findings, it is important that the authors explain what broader conclusions or questions can be drawn from this, and thereby show the relevance of their research for broader contexts.

Reviewer #2: A One Health approach with multi-sectoral and transdisciplinary collaborations, to zoonotic and reverse zoonosis prevention is urgently needed in this area as highlighted in this small semi-structured questionnaire FGD study. Currently strengthening of multisectoral collaboration for Mpox and anthrax prevention is occurring in the Kanungu district due to the location near DRC and cross-border interaction between school children. Training of approximately 200 school managers with IEC materials (posters and flyers) and a collaborative response action plan is centred on child-to-child strategy of empowering learners to share health message with peers, families and communities.

Further validation studies are required to evaluate the reason why some participants had awareness of reverse zoonosis transmission of disease from humans to gorillas. It is not evidence based to state that this reverse zoonosis awareness was due to targeted educational gorilla health programs surrounding BINP. A scoping review of all current educational programs and training relating to zoonosis and reverse zoonosis prevention and One Health practices is needed.

The recommendations need to be re-written in light of the reviewer’s comments. A One Health approach for co-developed culturally sensitive and inclusive community evidence-based programs for health system strengthening across human, animal/wildlife and ecosystem is needed. Additionally continuing to invest in long-term community livelihood projects that support poverty alleviation and improving the health of livestock and wildlife is recommended.

Reviewer #3: Overall, this is an interesting study reporting on the knowledge of rural and urban communities in southwestern Uganda on the possibility of transmission of infectious diseases between wild animals and humans. The manuscript is mostly well written, clear, and concise. Appropriate conclusions and policy recommendations have been made that are backed up by the findings.

PLOS authors have the option to publish the peer review history of their article (what does this mean? ). If published, this will include your full peer review and any attached files.

**Do you want your identity to be public for this peer review?** For information about this choice, including consent withdrawal, please see our Privacy Policy .

Reviewer #1: No

Reviewer #2: **Yes: ** Dr Andrea Britton

Reviewer #3: No

**Figure resubmission:**

**Reproducibility:**



---

## [Decision Letter · Decision Letter 1]

3 Aug 2025

Assessment of exposure to zoonoses and perceptions of zoonotic transmission surrounding the Bwindi Impenetrable Forest, Uganda

Dear Dr. Haven,

Thank you for submitting your manuscript to PLOS Neglected Tropical Diseases. After careful consideration, we feel that it has merit but does not fully meet PLOS Neglected Tropical Diseases's publication criteria as it currently stands. Therefore, we invite you to submit a revised version of the manuscript that addresses the points raised during the review process.

Please submit your revised manuscript within 60 days Sep 02 2025 11:59PM. If you will need more time than this to complete your revisions, please reply to this message or contact the journal office at plosntds@plos.org. Please include the following items when submitting your revised manuscript:

We look forward to receiving your revised manuscript.

Kind regards,

Philip P. Mshelbwala

Academic Editor

Victoria Brookes

Section Editor

Shaden Kamhawi

co-Editor-in-Chief

Paul Brindley

co-Editor-in-Chief

**Additional Editor Comments :**

Reviewers raised serious methodological concerns regarding the approach used in this study, as well as issues related to previously provided comments that remain unaddressed. We therefore invite the authors to revise their manuscript, taking these concerns into account, and to provide a detailed point-by-point response to all reviewer comments.

For the quantitative component, the authors should clearly describe the data used in the analysis, explain how the outcome variable (zoonotic understanding) was derived, and provide more detail on the analytical methods and results, including how model performance was assessed. I'm not sure what you meant by "while minimizing the Akaike Information Criterion (AIC)." Do you mean that you selected the model with the lowest AIC value?

**Journal Requirements:**

At this stage, the following Authors/Authors require contributions: Scott Kellermann, Evan Andrew Rusoja, Micheal Wilkes, Nicole R Gardner, and Tierra Smiley Evans. Please ensure that the full contributions of each author are acknowledged in the "Add/Edit/Remove Authors" section of our submission form.

2) We have noticed that you referred to "Supplementary material Appendix 1" on page 5 in your manuscript. However, there is no corresponding file uploaded to the submission. Please upload it as a separate file with the item type 'Supporting Information'. Please add a full list of legends for your Supporting Information files after the references list.

3) Please ensure that the Data Availability Statement mentioned in the manuscript matches the one provided in the online submission form.

**Comments to the Authors:**

**Please note that one of the reviews is uploaded as an attachment.**

**Reviewers' Comments:**

Reviewer's Responses to Questions

**Key Review Criteria Required for Acceptance?**

**Methods**

-Are the objectives of the study clearly articulated with a clear testable hypothesis stated?

-Is the study design appropriate to address the stated objectives?

-Is the population clearly described and appropriate for the hypothesis being tested?

-Is the sample size sufficient to ensure adequate power to address the hypothesis being tested?

-Were correct statistical analysis used to support conclusions?

-Are there concerns about ethical or regulatory requirements being met?

Reviewer #1: Please see the attached document

Reviewer #2: The title of the manuscript could be more descriptive of the study. The reviewer suggests a title including “ Study of zoonoses and anthropozoonosis knowledge and perception of transmission activities in rural and urban tribal communities surrounding the Bwindi Impenetrable Forest in Uganda”.

The mixed-methods study methodology section is much clearer. The reviewer acknowledges updates:

•No participants withdraw from the study line

•Inclusion of study being conducted near the Bwindi Impenetrable Forest in Uganda

•Importance of One Health approach in zoonotic disease control

Reviewer #4: Overall the results presented match the goals of the analysis. Here are some few recommendations: I will advise authors to consider some potential limitations of the snowball technique used. Include some limitations of using this sampling method in your discussion. As it is non randomised, what are the potential limitations that would impact your data and results? This was not discussed.

Line 171 - was this selection done before the snowball sampling was done? If after, was sample size sufficient for the investigation? You stated that target population was individuals with direct or indirect contact with wild animals, What were the other group/s excluded and was there a reason? Please make this clear

**Results**

-Does the analysis presented match the analysis plan?

-Are the results clearly and completely presented?

-Are the figures (Tables, Images) of sufficient quality for clarity?

Reviewer #1: Please see the attached document

Reviewer #2: The aim of the study was to better understand communities contacts with wild and domestic animals and their knowledge of potential disease risks associated with interactions around the Bwindi Impenetrable National Park, a known foci for spillover events.

Knowledge is skills and information which is objective while awareness or perception is interchangeable and is experiences of the observer which helps a person understand better. Knowledge is used when the information is factual like figure 2 summarizes community knowledge about transmission. Awareness is associated with perceptions people have and exhibit.

It is still unclear to the reviewer how overall zoonotic awareness in Table 2 has been calculated.

Reviewer #4: Figure 2- figure captions should be at the bottom.

Table 2- state if this is a univariable analysis. Parenthesis missing in contact with vermins.

Table 3 and line 289, if you want to include age as a factor, I will advise you flip this round and use <35 as associated, and put the OR, instead of >35 which does not show association in the tables.

**Conclusions**

-Are the conclusions supported by the data presented?

-Are the limitations of analysis clearly described?

-Do the authors discuss how these data can be helpful to advance our understanding of the topic under study?

-Is public health relevance addressed?

Reviewer #1: Please see the attached document

Reviewer #2: The reviewer acknowledges the significantly improved conclusion emphasising the need for One Health strategies at national and international levels.

Reviewer #4: Line 393 – “We found that only 1/3 of the respondents were aware of zoonotic diseases, with rural communities having higher awareness” what do you suggest could explain this? If your data cannot explain this, would you suggest that future research is needed to understand this….?

Line 419- I understand that this maybe beyond the scope of this study, however, it does come up in your MLR that rural has more awareness, you need to provide at least some reasons why you believe this is so and then advice that further research can be done….

Line 448 - It will be good to include any suggestions here on the type of study that will be relevant

**Editorial and Data Presentation Modifications?**

Reviewer #1: Please see the attached document

Reviewer #2: (No Response)

Reviewer #4: I have recommended some modifications that I believe will enhance the clarity of this publication. I will recommend minor revisions

**Summary and General Comments**

Reviewer #1: Please see the attached document

Reviewer #2: The revised manuscript is much improved and most reviewer comments were addressed.

Reviewer #4: Overall, the goals of the study was answered however, I have given a few points that the authors should consider. “the goal of this study was to better understand communities’ contacts with wild and domestic animals and their knowledge of potential disease risks associated with these interactions”. Add if this for policy purposes or for awareness raising somewhere in the discussion.

Some strengths of the research includes that the translation was reviewed by research team who were fluent in the local languages to ensure that meaning of words in the questionnaire was retained.

The methodology used, mixed model was appropriate, and odds ratio including confidence intervals were reported, I have advised risk factors such as decreasing age be reported rather than increasing age which seem like a protective factor.

It is an important piece of research that helps to provide some information on the understanding of zoonoses in the study area. I recommend for publication after revision of the minor corrections.

PLOS authors have the option to publish the peer review history of their article (what does this mean? ). If published, this will include your full peer review and any attached files.

**Do you want your identity to be public for this peer review?** For information about this choice, including consent withdrawal, please see our Privacy Policy .

Reviewer #1: No

Reviewer #2: **Yes: ** Andrea Britton

Reviewer #4: **Yes: ** Bibiana Zirra Shallangwa

**Figure resubmission:**

**Reproducibility:**



---

## [Editor Report · Decision Letter 2]

30 Oct 2025

Dear Mr Haven,

We are pleased to inform you that your manuscript 'Assessment of exposure to zoonoses and perceptions of zoonotic transmission surrounding the Bwindi Impenetrable Forest, Uganda' has been provisionally accepted for publication in PLOS Neglected Tropical Diseases.

Best regards,

Philip P. Mshelbwala

Academic Editor

Shaden Kamhawi

co-Editor-in-Chief

Paul Brindley

co-Editor-in-Chief

---

## [Editor Report · Acceptance letter]

Dear Mr Haven,

We are delighted to inform you that your manuscript, " 

Assessment of exposure to zoonoses and perceptions of zoonotic transmission surrounding the Bwindi Impenetrable Forest, Uganda," has been formally accepted for publication in PLOS Neglected Tropical Diseases.

Best regards,

Shaden Kamhawi

co-Editor-in-Chief

Paul Brindley

co-Editor-in-Chief
